# Structure of a low-population intermediate state in the release of an enzyme product

**Alfonso De Simone[1], Francesco A Aprile[2], Anne Dhulesia[2], Christopher M Dobson[2], Michele Vendruscolo[2]***

[1]Department of Life Sciences, Imperial College London, London, United Kingdom;
[2]Department of Chemistry, University of Cambridge, Cambridge, United Kingdom

**Abstract** Enzymes can increase the rate of biomolecular reactions by several orders of magnitude. Although the steps of substrate capture and product release are essential in the enzymatic process, complete atomic-level descriptions of these steps are difficult to obtain because of the transient nature of the intermediate conformations, which makes them largely inaccessible to standard structure determination methods. We describe here the determination of the structure of a low-population intermediate in the product release process by human lysozyme through a combination of NMR spectroscopy and molecular dynamics simulations. We validate this structure by rationally designing two mutations, the first engineered to destabilise the intermediate and the second to stabilise it, thus slowing down or speeding up, respectively, product release. These results illustrate how product release by an enzyme can be facilitated by the presence of a metastable intermediate with transient weak interactions between the enzyme and product.

***For correspondence:** mv245@
cam.ac.uk

**Competing interests:** The authors declare that no competing interests exist.

**Reviewing editor**: John Kuriyan, Howard Hughes Medical Institute, University of California, Berkeley, United States

## Introduction

As it is becoming increasingly clear that proteins populate a variety of 'intermediate' states during their function (*Dobson, 2003*; *Sekhar and Kay, 2013*), it is essential to determine the structures of such states in addition to defining the native conformations. Protein intermediates are involved in folding, misfolding, and aggregation processes, as well as in events associated with molecular recognition, catalysis, and allostery (*Dobson, 2003*; *Sekhar and Kay, 2013*; *Tzeng and Kalodimos, 2013*). These species are transient in nature and as such they have been difficult to characterise. Nuclear magnetic resonance (NMR) spectroscopy has emerged in this context as a powerful technique to define such states as exemplified by the characterisation of the structures of species involved in folding (*Korzhnev et al., 2010*), molecular recognition (*Tang et al., 2006*), and aggregation (*Neudecker et al., 2012*).

In the present paper, we describe a study of the mechanism involved in the process by which an enzyme releases its products. This is one of the three major steps in an enzymatic catalysis process (*Fersht, 1999*). In the first step, the enzyme forms a complex with the substrate. In the second step, the transition state of the reaction is reached within the favourable environment provided by the catalytic site enabling the conversion of the substrate into product. In the third step, which is often rate-limiting, the product is released and the enzyme returns to its original state. Each of these steps is usually rather complex and involves reaction intermediates, which are transient in nature and difficult to characterise.

In order to investigate the third step, we have studied here lysozyme, the first enzyme to be crystallised (*Blake et al., 1965*), and whose structural properties have been characterised in great detail (*Blake et al., 1965*; *Phillips, 1967*; *Artymiuk and Blake, 1981*; *Radford et al., 1992*). The native

**eLife digest** Enzymes are proteins that catalyse biochemical reactions. They bind to their target molecules—known as substrates—and help to change them to make 'products'. Afterwards, the products are released and the enzymes are free to bind to the next molecules. To perform this activity, an enzyme can change its structure several times, but it has been challenging to characterise the intermediate shapes because of their transient nature.

De Simone et al. took advantage of a technique called nuclear magnetic resonance spectroscopy to get a better look at the structures adopted by the human enzyme lysozyme. This enzyme helps to protect us from bacterial infections because it breaks the links between peptidoglycan molecules, which make up the wall that surrounds bacterial cells.

The experiments show that two 'arms' in the lysozyme structure move to create an intermediate shape during the final step—the release of the product—in the chemical reaction. This type of flexibility gives the enzyme the ability to tightly bind the peptidoglycan at the start and to let go of the product when the chemical reaction is complete.

Next, to confirm their findings, De Simone et al. examined what happened when they introduced particular mutations in the gene that makes lysozyme. The first mutation was meant to destabilise the intermediate shape of the enzyme, which resulted in the enzyme binding more tightly to the peptidoglycan in the final step and releasing it more slowly. A second mutation was made to stabilize the structure of the intermediate shape, which, as expected, allowed lysozyme to release the peptidoglycan more quickly.

De Simone et al.'s findings explain how intermediate shapes can be involved in the release of the product from lysozyme and other enzymes. The next challenges will be to characterise the structure of the intermediate shape that binds to the substrate and, more generally, to extend this type of approach to other enzymes.

structure of this enzyme is divided into a α domain (residues 1 to 38, and 86 to 130) and β domain (residues 39 to 85), containing primarily α-helical and β-sheet secondary structures, respectively (*Blake et al., 1965*; *Phillips, 1967*; *Artymiuk and Blake, 1981*; *Radford et al., 1992*). This enzyme degrades bacterial cell walls by catalysing the hydrolysis of the 1,4-β-linkages of the cell wall peptidoglycans, with a reaction that has been the object of intense scrutiny (*Chipman and Sharon, 1969*; *Warshel and Levitt, 1976*; *Post et al., 1986*; *Vocadlo et al., 2001*). According to the mechanism originally proposed by Phillips on the basis of his structure (*Phillips, 1967*), lysozyme binds to a peptidoglycan molecule in the binding site within the cleft between its two domains thus causing the substrate to adopt a strained conformation similar to that of the transition state of the hydrolysis. Here, we study the product release process. To this end, we used a well-characterised oligosaccharide product having an inhibitory effect on the enzyme, N,N',N"-triacetylchitotriose (triNAG) (*Turner and Howell, 1995*), which has been frequently used for studying lysozyme–product interactions (*Post et al., 1986*).

## Results and discussion

In order to define the structural populations of human lysozyme in the presence and in the absence of an inhibitor, we measured $^{15}$N-$^1$H residual dipolar couplings (RDCs) (*Tjandra and Bax, 1997*; *Tolman et al., 1997*) in the ligand-free and ligand-bound states (see 'Materials and methods') and used them as structural restraints in molecular dynamics simulations (*De Simone et al., 2011*; *Montalvao et al., 2011*). In this way, we determined two ensembles of structures of the enzyme representing, respectively, the ligand-free and the ligand-bound states of this protein. Our results indicate that large-scale concerted motions between the α and β domains of the enzyme generate an intermediate state involved in the release of the product.

The use of RDCs as structural restraints assists the conformational sampling in molecular dynamics simulations in order to estimate the free-energy landscape of a protein, as recently shown with hen lysozyme for which a large body of experimental data were used for validation purposes (*De Simone et al., 2013b*). This approach enables the translation of the experimental measurements into structures according to the principle of maximum entropy (*Pitera and Chodera, 2012*; *Cavalli et al., 2013*; *Roux and Weare, 2013*). A number of methods to employ NMR measurements of RDCs for the

characterisation of the structure and dynamics of proteins have been proposed (*Clore and Schwieters, 2004b*; *Bouvignies et al., 2006*; *Lange et al., 2008*). Since these approaches have generally been used to assess dynamic events of relatively small amplitude, it was necessary to derive a means of extending these methods to enable the description of the large conformational interconversions associated with the function of many protein molecules.

The strategy that we have used for this purpose relies on the ability to extract from time and spatially averaged data the contributions to the experimental observables that come from the low-population states present as a result of conformational fluctuations. Intermediate states determined in this way have been already described using paramagnetic resonance enhancement (PRE) (*Tang et al., 2006*) and RDC (*De Simone et al., 2013a*; *De Simone et al., 2013b*) measurements, complementing approaches in which NMR parameters specific for these states are obtained directly, in particular by relaxation–dispersion methods (*Korzhnev et al., 2004*, *2010*; *Bouvignies et al., 2011*; *Neudecker et al., 2012*). By applying this approach using RDCs to human lysozyme, we generated a structural ensemble representing the free state of this enzyme that reveals large breathing motions between the α and β domains (*Figure 1A* and *Figure 1—figure supplements 1,2*). This motion, which influences the mutual orientation of the two domains thereby altering the structure of the catalytic pocket at their interface, can be accounted for by defining a 'breathing' angle θ between the α-domain, the hinge region and the β-domain (*De Simone et al., 2013b*) (*Figure 1—figure supplement 3*).

In order to compare the free and bound states of the enzyme, we measured the RDCs also in the bound state (*Figure 1A*, *Figure 1—figures supplements 1 and 2*). While in the free state the free-energy landscape is characterised by a single basin (*Figure 1A*), in the bound state a second local minimum appears (*Figure 1B*). This change, which reflects the differences in the RDC data of the free and bound states, corresponds to a wider distribution of values of the θ angle in the bound state compared with the free state (*Figure 1* and *Figure 1—figure supplement 4*). The bound state ensemble was validated using NOEs, J-couplings, chemical shifts and RDCs (*Figure 1—figure supplement 5*). The relative populations of the ground and excited states are comparable to those that we have studied previously using the method adopted here (*De Simone et al., 2009*; *De Simone et al., 2011*; *De Simone et al., 2013a*; *De Simone et al., 2013b*).

Having in mind the release of the product, we designate the global free energy minimum observed in this study as the 'locked state' (i.e., release incompetent), which is centred at θ values of about 58° and Cα-RMSD values of about 0.9 Å from the X-ray structure of the complex (calculated by considering secondary structure elements only), and the other free energy minimum, which has about a 13% population under the conditions of our experiments, defined as the 'unlocked state' (i.e., release competent, *Figure 1B*). The unlocked state is a compact conformation that differs from the locked state by a global motion in which the α and β subunits become closer, with a θ value of about 49° in the centre of the basin. This motion generates particularly distorted structures with global RMSD values of about 1.5 Å from the X-ray structure. The angle θ provides a simple and effective reaction coordinate to describe the effect of triNAG binding on the energy landscape of human lysozyme (*Figure 2A*), which clearly illustrates how the protein is able to explore closed conformations (i.e., θ < 50°) upon ligand binding.

In the structural ensemble representing the complex between human lysozyme and triNAG, unlocked conformations are characterised by less favourable intermolecular Coulomb and van der Waals interactions than those found in the locked state (*Figure 2B–E*). A comparison between the locked and unlocked conformations indicates that this difference corresponds to specific interactions between the substrate and the binding pocket (*Figure 2D*), which include hydrophobic interactions between a methyl group of triNAG and the side-chain of W109, as well as hydrogen bonds between the ligand and the main chain amide group of N60 and the side chains of W64 and Q104. These interactions are present in essentially all the structures in the ensemble representing the locked state, while they are absent in the structures in the ensemble of the unlocked state. Indeed, because of a partial displacement of the ligand from the binding pocket, the unlocked conformations lose the tight interactions that are stabilised in the locked state and gain new interactions on the external surface of the protein. These interactions, which mainly involve hydrogen bonds between donor and acceptor groups from the ligand and the protein surface, are highly variable and heterogeneous in the unlocked conformations.

Overall, this analysis of the structural ensembles of human lysozyme suggests that, as a consequence of a concerted conformational transition, the enzyme explores conformations in which the specific and tight intermolecular interactions with the substrate in its locked state are largely lost in favour of the formation of weak and non-specific interactions in its unlocked state. This transition is

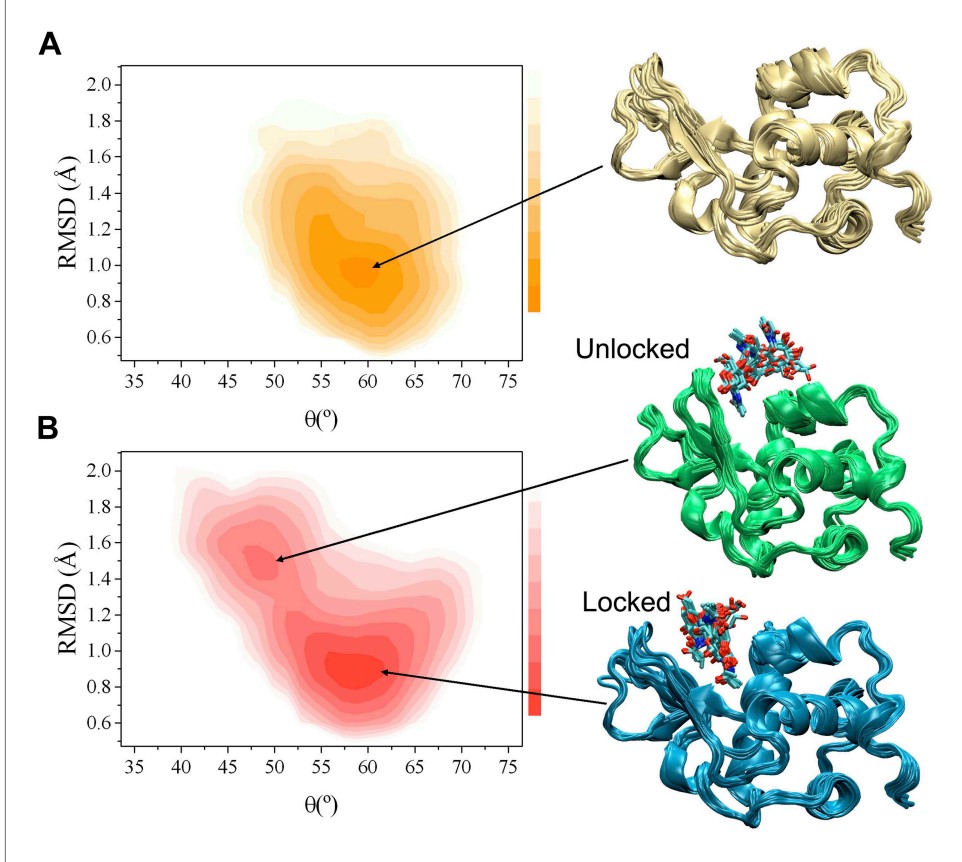

**Figure 1**. Comparison of the free-energy landscapes of human lysozyme in the free state (**A**) and in the bound state with triNAG (**B**). The bound state exhibits a ground state (the 'locked state') and an additional local minimum with about 13% population (the 'unlocked state'), which represents an intermediate in the release of the product of the enzymatic reaction. Free-energy landscapes are shown as function of the 'breathing' angle θ and of the RMSD from the X-ray structure, which was calculated on the Cα atoms by including secondary structure regions only, of a human lysozyme variant in complex with triNAG (PDB code 1BB5); free-energy landscapes were obtained as $-k_B T \ln H(θ, RMSD)$, where $H(θ, RMSD)$ is the number of times conformations with specific θ and RMSD values was sampled during the simulations (*De Simone et al., 2013b*).

The following figure supplements are available for figure 1:

**Figure supplement 1**. Assignments of the $^1H$-$^{15}N$ HSQC spectra of the free and triNAG-bound states of human lysozyme.

**Figure supplement 2**. Extracts of $^1H$-$^{15}N$ HSQC spectra showing the titration of triNAG to human lysozyme for selected residues showing significant chemical shift changes upon binding.

**Figure supplement 3**. Illustration of the breathing angle θ of lysozyme (*De Simone et al., 2013b*), which accounts for the large-amplitude motion between the α-domain and β-domain of lysozyme and is computed from the centres of mass of Cα-atoms from three protein regions (*De Simone et al., 2013b*).

**Figure supplement 4**. (**A**, **B**) Experimentally measured $^{15}N$-$^1H$ residual dipolar couplings (RDCs) of human lysozyme in the free state (**A**) and the triNAG-bound state (**B**).

**Figure supplement 5**. Validation of the RDC-refined structural ensembles determined in this work representing the free and triNAG-bound states of human lysozyme.

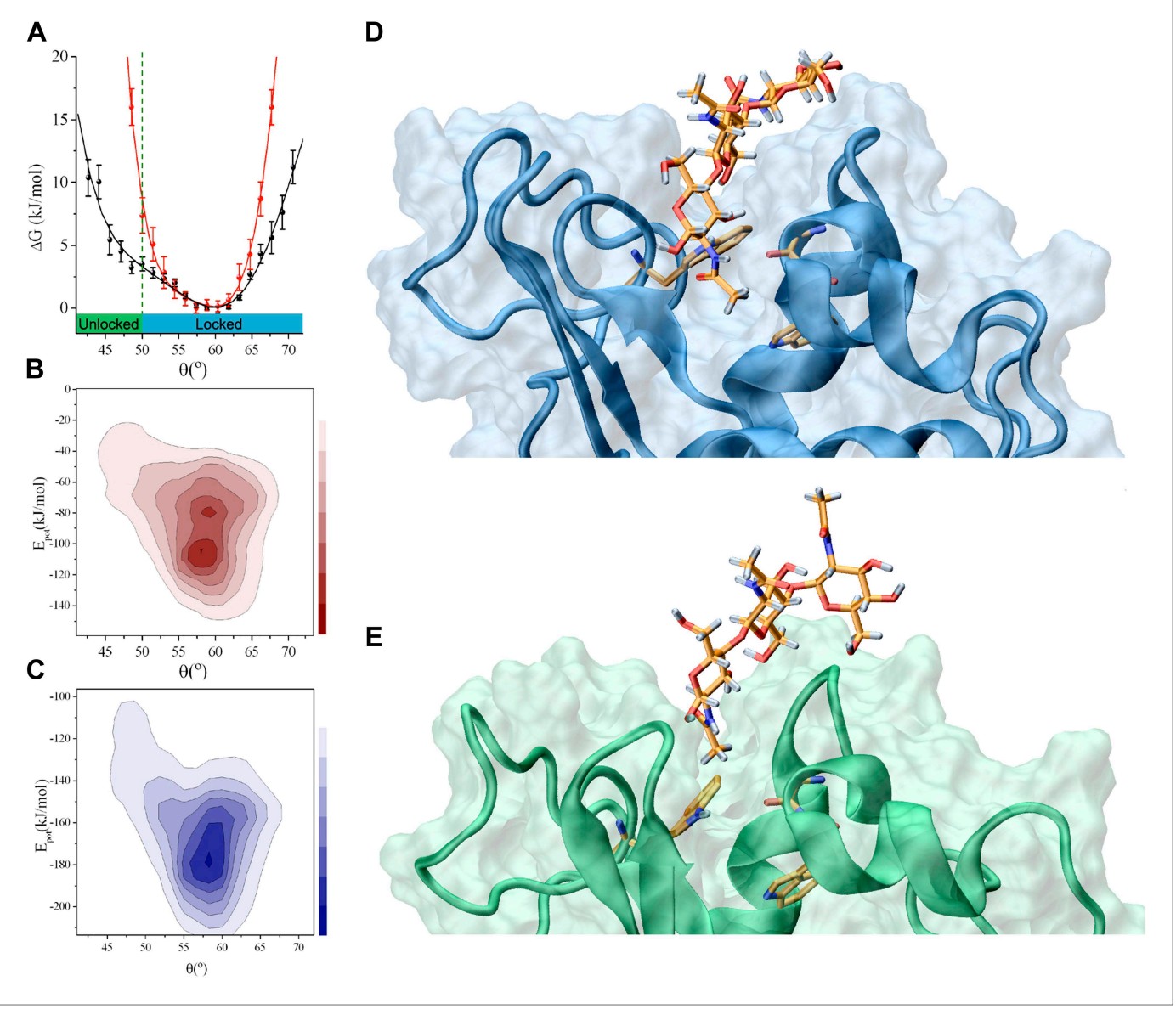

**Figure 2**. Analysis of the interactions that stabilise the intermediate state in the release of the product (the 'unlocked state'). (**A**) Free-energy landscape as a function of the angle θ. (**B**) Potential energy landscape, $E_{pot}$, of lysozyme in the free state; $E_{pot}$ represents the contribution of the force field used in the simulations, that is, the total force field without the RDC restraint term (see 'Materials and methods'). (**C**) Potential energy landscape, $E_{pot}$, of the lysozyme-triNAG bound state. (**D**) Structure of the 'locked state'. (**E**) Structure of the 'unlocked state'.

favoured by large-scale conformational motions in which the α and β domains become closer, thus suggesting that these motions are employed by the enzyme to modulate the affinity with the ligand. The unlocked state therefore represents an intermediate state for product release. In this view (*Figure 3*), the enzyme product complex (EP) populates transiently an intermediate state (EP*) that favours the release of the product (E + P). Thus, the analysis of the structural ensembles that we have determined provides evidence that large-scale conformational transitions are employed by enzymes along their catalytic cycles including key events in the product release step, which often represents the rate-limiting step that governs the turnover of the enzyme. The difficulty for enzymes to release the products can arise from the fact that typically the latter have similar physico-chemical characteristics to the substrates and therefore maintain a significant affinity for the enzyme.

To test the possibility that the structure that we have determined of the unlocked state represents an intermediate state in the release of the product, we compared in detail the energetic contributions

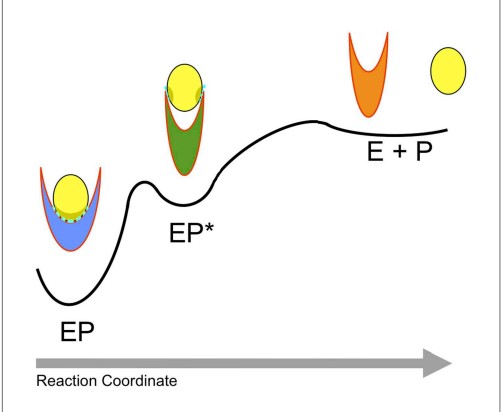

**Figure 3**. Schematic illustration of the process of product release. The product (P) is released by the enzyme (E) in a process that begins in the ground (or 'locked') state of the complex (EP), visits a metastable (or 'unlocked') intermediate state (EP*) and reaches the unbound state (E + P). The interactions in the 'locked state' (EP) and in the 'unlocked state' (EP*) are shown in light blue.

that stabilise the locked and unlocked states and identified a stabilising hydrogen bond that can be formed only in the unlocked state (*Figure 4A*), which involves side-chain atoms of residue N44 in the β-domain and residue E35 in the α-domain. These two residues are too far away from each other in the locked state to form a hydrogen bond (*Figure 4B*), and therefore this interaction is characteristic only of the unlocked conformation. This finding suggests that this hydrogen bond provides a specific interaction by which one could selectively target the stability of the EP* intermediate. To this end, we rationally designed the N44A single-amino acid mutation to verify if the ability of the enzyme to release the products is indeed altered by preventing the formation of the hydrogen bond that stabilises the unlocked state.

Comparison of the $^1$H-$^{15}$N-HSQC spectra of the wild-type and the N44A variant of human lysozyme shows that the mutation does not affect the structural properties of the mutant in the native state (*Figure 4—figure supplement 1*). This result was expected as the mutation does not modify interactions present in the native state but was designed explicitly to perturb a hydrogen bond in the intermediate state. The ability of the N44A variant to release triNAG from its bound state was assessed by surface plasmon resonance (SPR) experiments (*Figure 4C*). By using a double-referenced single chain model (see 'Materials and methods'), we estimated that the destabilisation of the intermediate in the N44A mutant reduces the $k_{off}$ by a factor 3, while changing the $K_d$ by a factor 1/3. The variation of a factor 2 of the $k_{on}$ suggests that the pathways of capture and release are not completely distinct and thus perturbing the pathway for release affects in part also that of capture. These results are consistent with our prediction that the N44A mutation affects the stability of the EP* intermediate, that is, of the unlocked state. Finally, we tested the catalytic efficiency of the wild-type and N44A mutant by using a cellular assay (see 'Materials and methods'), and compared these results with those obtained for an inactive mutational variant lacking the catalytic residue (E35D). The results (*Figure 4D*) show that the N44A variant has an intermediate activity between wild type and the totally inactive control E35D variant, which is again consistent with the prediction that the mutation of asparagine to alanine of residue 44 would reduce the efficiency of the product release in such a way to affect the catalytic activity of the enzyme. Finally, to verify that the N44A modified the free-energy landscape of lysozyme by reducing the population of the unlocked state, we performed $^{15}$N-$^1$H RDC measurements on the N44A mutant and carried out restrained molecular dynamics simulations to determine its free-energy landscape. The results (see *Figure 4— figure supplement 2A*) demonstrate that the unlocked state is not appreciably populated in the N44A mutant.

To further validate the conclusion that the structure that we have determined of the unlocked state represents a release intermediate, we designed a second mutational variant to stabilise the unlocked state, rather than destabilising it as the N44A mutation. In the new mutant, N46Q/V110Q, a strong glutamine–glutamine interaction is inserted with the purpose to stabilise the 'unlocked' state in its conformation (*Figure 4—figure supplement 3A*). We have verified the folding of the mutant by NMR (*Figure 4—figure supplement 3B*) and measured the binding constants of the ligand for the unlocked state by SPR to show that it corresponds to a weaker binding affinity (*Figure 4—figure supplement 3C*). While the $K_d$ of the wild type is about 9 μM, the $K_d$ of the N46Q/V110Q mutant is high almost beyond detection, indicating that the mutant essentially does not bind the substrate. These experimentally measured binding constants are consistent with the observation that, considering that the free energy of the free state is the same, the binding free energy of the locked state is larger than that of the unlocked state because the free energy of the former is lower than that of the latter (*Figure 1*).

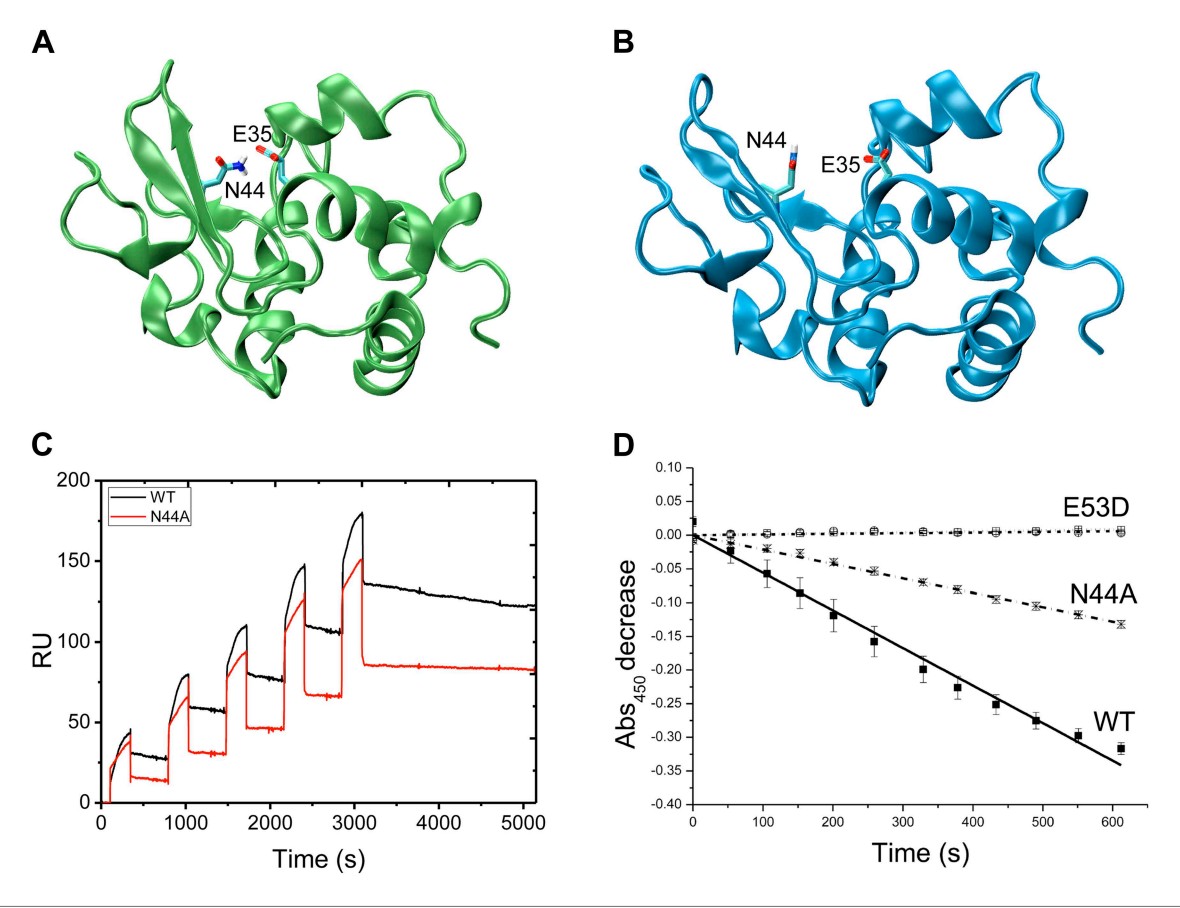

**Figure 4**. Experimental assessment of the role of the intermediate state determined in this work in the product release process. (**A**) Structure of the unlocked state illustrating the hydrogen bond between side chains of N44 and E35 that stabilises this intermediate species. (**B**) The hydrogen bond is not formed in the locked state because N44 and E35 are too far apart. (**C**) The N44A variant, which lacks the hydrogen bond donor, is unable to form this hydrogen bond, thus destabilising the intermediate state and inhibiting the release of the product. The decrease of the ability of the N44A mutant to release triNAG has been assessed by surface plasmon resonance (SPR) experiments. (**D**) Cellular assay of lysozyme activity. The N44A variant has an intermediate activity between wild type and the control E35D variant.

The following figure supplements are available for figure 4:

**Figure supplement 1**. Comparison of the $^1$H-$^{15}$N HSQC spectra of WT (black) and N44A mutant (red).

**Figure supplement 2**. Comparison of the free-energy landscapes of wild-type (red) and N44A mutant (black) lysozyme.

**Figure supplement 3**. Study of the N46Q/V110Q mutant.

## Conclusions

We have presented the atomic resolution structure of an intermediate associated with the product release in an enzymatic reaction. We have validated this structure by identifying a distinctive structural characteristic of this state, a transient hydrogen bond between the side-chains of residues N44 and E35. As this interaction stabilises the intermediate state but not the ground state, we introduced a mutational variant (N44A) that, by removing specifically the hydrogen bond, reduces the stability of the intermediate state but not that of the ground state and thus inhibits the release process. Our results provide an illustration of the manner in which conformational fluctuations can play a central role in enzymatic reactions by creating low-population intermediate states that facilitate the challenging step of release of the products of the catalytic reaction.

# Materials and methods

## Sample preparation

Human lysozyme was expressed in *Pichia pastoris* and purified on an ion exchange column, as previously described (*Johnson and et al., 2005*). $^{15}$N ammonium sulfate and $^{13}$C methanol were used to $^{15}$N and $^{13}$C label the protein, respectively. NMR experiments were carried out using a 700 MHz spectrometer at 37°C in a buffer at pH 5.0 containing 20 mM potassium phosphate and 10% $D_2O$; the pH was re-adjusted after the addition of the protein. Protein concentrations were in the range of 200–350 μM. For the measurements of the bound state, N,N′,N"-triacetylchitotriose (triNAG) sugar was purchased from Sigma and dissolved in water to constitute a concentrated stock solution.

## Assignment of NMR spectra

For the assignment of the free state at pH 5.0 and 37°C, we used a previously published assignment (*Ohkubo et al., 1991*; *Hagan and et al., 2010*), which was confirmed using HNCA measurements, which was run with a spectral width of 1561 Hz and 68 points in the $^{15}$N dimension, and a spectral width of 5456 Hz and 64 points in the $^{13}$C dimension (*Grzesiek and Bax, 1992*). In total, 126 backbone amides were assigned in the $^1$H-$^{15}$N spectrum.

For the full assignment of human lysozyme bound to triNAG, we performed titrations of $^1$H-$^{15}$N HSQC spectra of a 200 μM sample of $^{15}$N human lysozyme, which were recorded using progressive concentration of the ligand (0, 0.3, 0.5, 0.8, 1.1, 1.6, 2.4, 3.1, 5.2, and 10 equivalents), allowing us to sample different points along the binding curve. HSQC spectra were recorded with a spectral width of 1621 Hz and 128 points in the $^{15}$N dimension (*Figure 1—figure supplement 2*). Additional information was obtained using HNCA and HNCACB experiments of a triNAG-saturated human lysozyme sample (*Grzesiek and Bax, 1992*; *Muhandiram and Kay, 1994*). The HNCA experiment was carried out with the same settings as for the free state (see above). The HNCACB experiment was carried out with a spectral width of 1561 Hz and 68 points in the $^{15}$N dimension and with a spectral width of 13,210 Hz and 72 points in the $^{13}$C dimension. These complementary data allowed us to obtain the full assignment of the $^1$H-$^{15}$N spectra (*Figure 1—figure supplement 1*).

## Residual dipolar coupling measurements

Residual dipolar couplings (RDCs) were measured by orienting the free and triNAG-bound states in two different bicelle solutions, neutral and charged (*Ottiger and Bax, 1998*; *Schwalbe and et al., 2001*). The neutral bicelle solution contained 5% wt/vol of a mixture of DMPC and DHPC (q = 2.9), whereas CTAB was used to create a positively charged solution of 10% wt/vol of the (DMPC:DHPC:CTAB) = (2.9:1:0.2) composition. Splitting of the $^2$H signal was recorded before and after the IPAP experiments, to ensure that alignment had remained constant during the course of the NMR experiment. IPAP experiments were recorded on the isotropic sample as well as on the two anisotropic samples (neutral and charged) (*Ottiger et al., 1998*). These experiments were performed using a spectral width of 2447 Hz with 320 points in the indirect $^{15}$N dimension for the in-phase (IP) or anti-phase (AP) spectra. J-couplings were extracted in each medium and RDCs were derived, discarding overlapping and poorly defined peaks. For the free state, we extracted 109 RDCs in the steric medium and 110 in the charged medium; 109 RDCs were extracted for the bound state, both for steric and charged media.

## $^3$J scalar coupling measurements

$^3$J HNHα couplings were obtained using HNHA experiments (*Vuister and Bax, 1993*), which were performed on the free and bound states using a 700 MHz spectrometer and a spectral width of 1454 Hz with 68 points in $^{15}$N and 9800 Hz with 72–80 points in the indirect $^1$H dimension. The $^3$J HNHα couplings were extracted using the ratio of intensities of cross- ($I_X$) and diagonal ($I_D$) peaks (*Kuboniwa et al., 1994*)

$$I_X/I_D = -\tan^2(2\pi\xi\,^3J) \tag{1}$$

with ξ = 13.05 ms.

Errors in the $^3$J HNHα coupling values were based either on a 5% uncertainty or on the noise level for cross-peaks with intensities below the RMS noise of the HNHA spectrum, estimated using Sparky (Goddard, T. D., and D. G. Kneller. SPARKY 3. University of California, San Francisco, 2004). Errors on

intensities were propagated according to *Equation (1)* to yield the error on $^3$J HNHα couplings. Residues with overlapping diagonal peaks were discarded, as well as glycine residues.

## Molecular dynamics simulations

As a starting structure for the ligand-free state, we used the crystal structure of human lysozyme at 1.9 Å resolution (PDB code 2ZIJ). For the bound state, we used the crystal structure of the human lysozyme A96L variant bound to triNAG at 1.8 Å resolution (PDB code 1BB5). This structure was modelled by mutating back residue 96 from L to A, as in the wild-type sequence. Molecular dynamics simulations were performed by using AMBER99SB with corrections on backbone (*Best and Hummer, 2009*) and side chains (*Lindorff-Larsen et al., 2010*) dihedral angles as the force field ($E^{FF}$) for the protein. triNAG was modelled using the GLYCAM06 force field (*Kirschner et al., 2008*). The protein and protein/triNAG systems were solvated using the TIP3P water model (*Jorgensen et al., 1983*). A time step of 2 fs was used together with LINCS constraints (*Hess, 2008*). Systems were energy minimised and equilibrated with positional restrained simulations of 20 ns, in which the heavy atoms of the protein and triNAG molecules were restrained to their Cartesian coordinates. For the free state, the resulting system box after equilibration was 5.55 × 6.16 × 5.56 nm$^3$, with 5698 water molecules for a total of 19,123 atoms. For the bound state, the resulting system box after equilibration was 6.15 × 5.62 × 5.99 nm$^3$, with 6131 water molecules for a total of 20,509 atoms.

The simulations were performed in the NPT ensemble by weak coupling the pressure and temperature with external baths. Temperature coupling was performed with the v-rescale method (*Bussi et al., 2007*) with a coupling constant of 0.1 ps. The pressure was kept constant using the Berendsen method (*Berendsen et al., 1984*), with a coupling constant of 1.0 ps and at a reference pressure of 1 bar. The isotropic compressibility value was set to 4.5 × 10$^{-5}$ bar$^{-1}$. Electrostatic interactions were treated by using the particle mesh Ewald method (*Essmann and et al., 1995*).

## Molecular dynamics simulations with RDC restraints

We used replica-averaged RDC restraints in molecular dynamics simulations (*De Simone et al., 2011*; *Montalvao et al., 2011*; *De Simone et al., 2013a*; *De Simone et al., 2013b*). This method has been tested for its ability to sample interdomain motions in proteins (*De Simone et al., 2011*; *De Simone et al., 2013b*), as well as in multiple conformational states in fast exchange in the NMR measurements (*De Simone et al., 2013a*). A recent study was carried to generate accurate ensembles of hen egg white lysozyme using RDC measured under the same conditions of the present work (*De Simone et al., 2013b*). This investigation has defined the sampling method that we have used here to characterise the conformational properties of lysozyme using RDC restraints. The accuracy of the resulting ensemble was benchmarked using a large variety of NMR observables, including eight sets of RDCs. Briefly, in this approach (*De Simone et al., 2011*; *De Simone et al., 2013b*), the structural information provided by RDC measurements is imposed to restrain the molecular dynamics simulations by adding a term, $E^{RDC}$, to a standard molecular mechanics force field, $E^{Pot}$:

$$E^{Tot} = E^{Pot} + E^{RDC}. \tag{2}$$

The resulting force field, $E^{Tot}$, is employed in the integration of the equations of motion. In this work, the restraint term, $E^{RDC}$, is given by (*De Simone et al., 2011*; *De Simone et al., 2013b*):

$$E^{RDC} = \alpha \sum_i \left( D^{exp} - D^{calc} \right)^2, \tag{3}$$

where $\alpha$ is the weight of the restraint term, and $D^{exp}$ and $D^{calc}$ are the experimental and calculated RDCs, respectively. The RDC of a given bond vector is calculated as (*De Simone et al., 2011*; *De Simone et al., 2013b*):

$$D^{calc} = \frac{1}{M} \sum_m D_m, \tag{4}$$

where $m$ runs over the $M$ replicas and $D_m$ is the RDC of replica $m$, which is given by:

$$D = D_{max} \sum_{ij} \left\langle A_{ij} \right\rangle \cos \varphi_i \cos \varphi_j, \tag{5}$$

where $\varphi_i$ and $\varphi_j$ are the angles between the internuclear vector and the molecular reference frame, the indices $i$ and $j$ run over the three Cartesian coordinates, $x$, $y$, and $z$, and $\langle A_{ij} \rangle$ is the *(i,j)* component of the alignment tensor.

The use of replica-averaged molecular dynamics simulations enables one to generate an ensemble of conformations compatible with the experimental data according to the maximum entropy principle (*Pitera and Chodera, 2012*; *Cavalli et al., 2013*; *Roux and Weare, 2013*), at least in the limit of large $M$ and $\alpha$. We have previously shown (*Cavalli et al., 2013*), however, that it is possible to effectively achieve this limit even if the values of $M$ and $\alpha$ remain relatively small and thus obtain conformational ensembles that provide a good agreement between experimental and calculated observables. Following these procedures, we used here $M = 16$ and for the weight, $\alpha$, we first carried out an initial equilibration simulation at 310 K, during which the agreement between the calculated and experimental data was allowed to converge by gradually raising $\alpha$ to the largest possible value that did not generate numerical instabilities. Subsequently, we performed a series of 50 cycles of simulated annealing between 310 and 500 K to sample the conformational space. Each cycle was carried out for a total of 250 ps (125,000 molecular dynamics steps). For each cycle, we collected 24,000 structures (1 per ps in the final 50 ps of the final 30 cycles of each of the 16 replicas). These structures were employed for the analyses reported in this study.

The alignment tensor is calculated from the shape and charge of the protein molecule using a procedure recently described (*Montalvao et al., 2011*). We adopted such an approach here rather than the more commonly used singular value decomposition (SVD) method (*Clore and Schwieters, 2004b*; *Clore and Schwieters, 2004a*) because in the presence of conformational fluctuations of relatively large amplitude, such as those exhibited by hen lysozyme, the SVD method, when used in combination with the replica-averaging procedure of *Equations 2–5*, is less effective in capturing the motions of a protein (*De Simone et al., 2013b*). The reason is that the SVD method does not necessarily provide the actual alignment tensor of a given structure but rather the alignment tensor that generates the RDC values in the closest agreement with the experimental ones and hence is less well suited in describing the specific differences between the structures considered in the averaging procedure in *Equation (3)* (*Montalvao et al., 2011*; *De Simone et al., 2013b*).

This structure-based method was used here to calculate the orientations of lysozyme in two alignment media, one steric (DMPC/DHPC) and one electrostatic (DMPC/DHPC/CTAB). The Q factors for the refined ensembles of the free and bound states of human lysozyme were 0.10 in both cases.

In addition to the previous extensive benchmarks performed on the structural ensembles of the hen egg white lysozyme (*De Simone et al., 2013a*), which were obtained using the same protocol employed in this work, we performed here a set of additional validations based on NMR measurements not used as restraints in the simulations and by comparing the resulting experimental values with those back-calculated from our ensemble of human lysozyme (*Figure 1—figure supplement 5*).

## Mutagenesis

N44A mutation and E35D or D53N (control mutations) were introduced into the pPIC9/HuLys wt by using *QuikChange* XL II mutagenesis kit (Qiagen, Venlo, The Netherlands). The pPIC9 plasmid containing the point mutations of HuLys cDNA was linearised by digestion with StuI. Transformation into *Pichia pastoris* GS115 was performed by using *Pichia* EasyComp Transformation Kit (Life Technologies), according to manufacturer's instructions. Cell colonies were screened for lysozyme expression level by quantifying by NuPAGE analysis the amount of lysozyme produced in 10-ml mini-cultures. Protein expression and purification were performed as previously reported (*Johnson and et al., 2005*). Protein purity exceeded 95% as estimated by NuPAGE analysis. Protein concentrations were determined by absorbance measurements at 280 nm using theoretical extinction coefficients calculated with Expasy ProtParam.

## Surface plasmon resonance

Surface plasmon resonance (SPR) experiments were performed using a Biacore 3000 system (GE Healthcare). CM5 sensor chip surfaces were activated by using an amine coupling kit (GE Healthcare). WT and N44A lysozyme variants were immobilised to the activated surfaces by amine coupling at a density of 2500–3000 resonance units (RU). Single chain kinetic experiments were performed at 25°C using a flow rate of 20 µl/min in 50 mM phosphate pH 6.2, 100 mM NaCl. Serial dilutions (200 µM, 100 µM, 50 µM, 25 µM, and 12.5 µM) of N,N′,N″-Triacetylchitotriose (Tri-NAG, Sigma Aldrich) were

sequentially injected every 700 s using a contact time of 250 s for each injection. Data fitting was performed with the single chain kinetic module provided with the Biaevaluation software (Biacore GE lifesciences).

## Cellular assay of lysozyme activity

Hydrolase activity assay was performed using *Micrococcus lysodeikticus* cells (Sigma Aldrich) as the substrate. Cells of *Micrococcus* were suspended at 0.3 mg/ml in 100 mM potassium phosphate, pH 6.2, shortly before the assay. The decrease of Absorbance at 450 nm was monitored at 25°C in the presence of 20 nM lysozyme variants.

## Acknowledgements

This work was supported by grants from EPSRC, Leverhulme Trust and EU (ADS), and BBSRC and Wellcome Trust (CMD and MV).

## Additional information

### Funding

| Funder | Author |
| --- | --- |
| Engineering and Physical Sciences Research Council | Alfonso De Simone |
| Leverhulme Trust | Alfonso De Simone |
| European Commission | Alfonso De Simone |
| Biotechnology and Biological Sciences Research Council | Christopher M Dobson, Michele Vendruscolo |
| Wellcome Trust | Christopher M Dobson, Michele Vendruscolo |

The funders had no role in study design, data collection and interpretation, or the decision to submit the work for publication.

### Author contributions

ADS, Conception and design, Acquisition of data, Analysis and interpretation of data, Drafting or revising the article; FAA, AD, Acquisition of data, Analysis and interpretation of data; CMD, MV, Conception and design, Analysis and interpretation of data, Drafting or revising the article

## Additional files

### Major datasets

The following previously published datasets were used:

| Author(s) | Year | Dataset title | Dataset ID and/or URL | Database, license, and accessibility information |
| --- | --- | --- | --- | --- |
| Shoyama Y, Tamada T, Nitta K, Kumagai I, Kuroki R, Koshiba T | 2009 | Crystal Structure of Human Lysozyme Expressed in E. coli | http://www.pdb.org/pdb/explore/explore.do?structureId=2ZIJ | Publicly available at RCSB Protein Data Bank. |
| Headley AG, Roe SM, Pearl LH | 2009 | Human lysozyme mutant a96l complexed with chitotriose | http://www.pdb.org/pdb/explore/explore.do?structureId=1BB5 | Publicly available at RCSB Protein Data Bank. |

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
