## [Decision Letter]

Thank you for sending your work entitled “Structure of a low-population
intermediate state in the release of an enzyme product” for consideration at
*eLife*. Your article has been handled by John Kuriyan (Senior editor
and Reviewing editor) and three additional reviewers.

The reviewers agree that this is a fine piece of work that is overall very well
performed and the conclusions logically drawn. The results further highlight the
importance of characterizing lowly populated conformational states. Nevertheless, the
reviewers are also in agreement that some additional data would be needed to make the
conclusions more compelling. We ask that you respond by email to the points raised
below, paying particular attention to the major points. Your response should tell us how
you could address these issues in a realistic timeframe. A final decision on your paper
will be made after the reviewers have considered your response.

Ensembles of structures of the enzyme lysozyme were determined by combining NMR-detected
RDCs and MD simulations to structurally characterize an intermediate species along the
enzymatic pathway. In particular, the paper analyzes NMR data using a method called
ensemble-average molecular dynamics simulations. The analysis was carried out on the
ligand-free and the ligand-bound states of the enzyme. The data show that the ground
state (∼87%) exists in the locked state (the one wherein the product is still
bound to the enzyme) and in equilibrium with an energetically excited state (populated
∼13% of the time), which allows product release (termed the unlocked state). The
two states are related by an opening/closure of the ligand-binding cleft by ∼10
degrees.

The unlocked state with the bound ligand is a compact conformation that differs from the
locked state by a global motion, with global RMSD values of about 1.5 Å relative to
the X-ray structure. The unlocked conformations lose the tight interactions that are
stabilized in the locked state and gain new interactions on the external surface of the
protein. These interactions, which mainly involve hydrogen bonds between donors and
acceptors groups from the ligand and the protein surface, are highly variable and
heterogeneous in the unlocked conformations. It is hypothesized that the unlocked state
represents an intermediate state that is critical for fast product release. The authors
then identified a stabilizing hydrogen bond that can be formed only in the unlocked
state (Figure 4), which involves side-chain
atoms of residue N44 E35. The ability of the N44A variant to release triNAG from its
bound state was assessed by SPR. The dissociation constants of the wild-type and the
N44A variant to be, respectively, 4.19 sec-1 and 1.31 sec-1. This provides a degree of
cross-validation that this state is in fact important for release.

Major points:

1) One weakness identified by the reviewers is that it is only a speculation that the
ligand is more loosely bound in the “unlocked” state. The authors provide
no firm basis for this assertion. The paper provides only one mutant to test the key
concept that there is an unlocked state. Based on the available information, it seems
that one should be able to design a mutant that stabilizes the unlocked state. If so,
then one could: (1) confirm the existence of the state directly by NMR, (2) measure the
binding constants (kinetics and equilibrium) of the ligand for the unlocked state by SPR
to show that it is indeed faster and with a weaker binding affinity. From a
computational point of view, it should be possible to calculate the binding free energy
of the ligand for the locked and unlocked state, and show that the affinity is reduced
in the latter. Doing this additional work is substantial, but it would really
demonstrate that the conclusions are correct.

2) It might be thought that the RDC approach for detecting rare states would be inferior
to other approaches, like relaxation dispersion or PREs. In the case of relaxation
dispersion one relies on the fact that the minor state is only transiently formed so
that its effective linewidth is very large. Thus, the exchange process gives rise to a
net increased linewidth in the ground state that can be detected. In a similar manner,
as long as the excited state brings the NMR probes close to the spin label, a PRE effect
is observed in the ground state. By contrast, however, RDCs contain information about
each state and it is unlikely that the rare state has RDCs that are so different
relative to the ground state, making this approach potentially less sensitive. Can the
authors comment on the sensitivity of their method in general to studies of rare
conformations? How much lower than 15% can one go and how robust are the states that are
determined?

3) In this regard, the authors do address the robustness with their N44A mutant and this
is an important control. It would be very interesting to record RDC on this mutant (free
form) and show that the same profile as in Figure 1 is observed. Second, it would then be of great interest to add the product
to the N44A mutant, measure RDC and repeat the calculations. One would predict that the
rare conformer would become less populated, less than 15%, and that would provide
further important proof of the methodology that jives with the functional (release)
studies that they perform. These controls are not particularly onerous but they would be
very convincing. At issue here to some extent is the reliability of RDCs for dynamics,
and there has been much debate in the literature about this (especially when one
considers the fact that methods for estimating RDCs from structure are only approximate,
especially for electrostatic alignment, or methods for dealing with the effects of
averaging between multi-states again can be complex). This work opens up an approach for
looking at rare states, and further evidence of utility is important.

4) Although not necessary, have the authors considered experiments to show that the
initial binding step of substrate is unaffected by the N44A mutation, as would be
predicted since the ground state is unaffected?

---

## [Author Response]

*1) One weakness identified by the reviewers is that it is only a speculation
that the ligand is more loosely bound in the “unlocked” state. The
authors provide no firm basis for this assertion. The paper provides only one mutant
to test the key concept that there is an unlocked state. Based on the available
information, it seems that one should be able to design a mutant that stabilizes the
unlocked state. If so, then one could: (1) confirm the existence of the state
directly by NMR, (2) measure the binding constants (kinetics and equilibrium) of the
ligand for the unlocked state by SPR to show that it is indeed faster and with a
weaker binding affinity. From a computational point of view, it should be possible to
calculate the binding free energy of the ligand for the locked and unlocked state,
and show that the affinity is reduced in the latter. Doing this additional work is
substantial, but it would really demonstrate that the conclusions are
correct*.

We are grateful for these key suggestions. To take account of them, we have used the
structure of the intermediate that we calculated to rationally design a new mutant meant
to stabilise the unlocked state. In the new mutant, N46Q/V110Q, a strong
glutamine-glutamine interaction is inserted with the purpose to effectively block the
‘unlocked’ state in its conformation (see Figure 4—figure supplement 3).

We have verified the folding of the new mutant by NMR (see Figure 4—figure supplement 3) and measured the binding
constants of the ligand for the unlocked state by SPR to show that it corresponds to a
weaker binding affinity (see Figure 4—figure supplement 3).Variantk_on_ (M^-1^s^-1^)St. Errk_off_ (s^-1^)St. ErrWT7516.7E-042E-05N46Q/V110Q13nd1.4E+04nd

While the Kd of the wild type is about 9 uM, the Kd of the N46Q/V110Q mutant is high
almost beyond detection, indicating that the mutant essentially does not bind the
substrate. These experimentally measured binding constants are consistent with the
observation that, considering that the free energy of the free state is the same, the
binding free energy of the locked state is larger than that of the unlocked state
because the free energy of the former is lower than that of the latter (Figure 1).

In order to validate the design procedure of the mutational variants with modified
stability of the unlocked state, we have performed new RDC measurements of the N44A
variant and used them to determine the corresponding free energy landscape (see Figure 4—figure supplement 2). We have not
repeated this NMR analysis for the N46Q/V110Q mutant since the measurements of the
binding constants provided already clear indications of the success of the design
procedure. If the editors, however, believe that we should perform the additional NMR
analysis also on the second mutant we will be happy to do so.

*2) It might be thought that the RDC approach for detecting rare states would be
inferior to other approaches, like relaxation dispersion or PREs. In the case of
relaxation dispersion one relies on the fact that the minor state is only transiently
formed so that its effective linewidth is very large. Thus, the exchange process
gives rise to a net increased linewidth in the ground state that can be detected. In
a similar manner, as long as the excited state brings the NMR probes close to the
spin label, a PRE effect is observed in the ground state. By contrast, however, RDCs
contain information about each state and it is unlikely that the rare state has RDCs
that are so different relative to the ground state, making this approach potentially
less sensitive. Can the authors comment on the sensitivity of their method in general
to studies of rare conformations? How much lower than 15% can one go and how robust
are the states that are determined*?

We recognise the importance of these considerations. In the revised version of the
manuscript we have addressed the issue of the sensitivity of the approach that we have
used, and briefly reviewed previous publications in which we have demonstrated that it
can be employed to characterise populations as low as those studied here (De Simone et
al., JACS 131, 3810-3811, 2009; J. Chem., Theor. Comput., 7, 4189-4195, 2011;
Biochemistry 52, 6684-6694, 2013; Biochemistry 52, 6480-6486, 2013).

*3) In this regard, the authors do address the robustness with their N44A mutant
and this is an important control. It would be very interesting to record RDC on this
mutant (free form) and show that the same profile as in*
Figure 1
*is observed. Second, it would then be of great interest to add the product to
the N44A mutant, measure RDC and repeat the calculations. One would predict that the
rare conformer would become less populated, less than 15%, and that would provide
further important proof of the methodology that jives with the functional (release)
studies that they perform. These controls are not particularly onerous but they would
be very convincing. At issue here to some extent is the reliability of RDCs for
dynamics, and there has been much debate in the literature about this (especially
when one considers the fact that methods for estimating RDCs from structure are only
approximate, especially for electrostatic alignment, or methods for dealing with the
effects of averaging between multi-states again can be complex). This work opens up
an approach for looking at rare states, and further evidence of utility is
important*.

To take account of this point, we performed additional 1H-15N RDC measurements on the
N44A mutant and carried out new restrained molecular dynamics simulations to determine
its free energy landscape. The new results (see Figure 4—figure supplement 2) demonstrate that the unlocked state is not
populated in the N44A mutant, as expected from its rational design and the measurement
of the binding constants reported in the original version of the manuscript.

*4) Although not necessary, have the authors considered experiments to show that
the initial binding step of substrate is unaffected by the N44A mutation, as would be
predicted since the ground state is unaffected*?

We have checked whether the binding step is affected by the N44A mutation.Variantk_on_ (M^-1^s^-1^)St. Errk_off_ (s^-1^)St. ErrWT7516.7E-042E-05N44A38.70.72.5E-043E-05

The destabilisation of the intermediate in the N44A mutant brings down the
k_off_ by a factor 3, while changing the K_d_ by a factor 1/3. The
k_on_ varies by a factor 2, indicating that the pathways of capture and
release are not completely distinct and thus perturbing the pathway for release affects
in part also that of capture.